# Integration between Novel Imaging Technologies and Modern Radiotherapy Techniques: How the Eye Drove the Chisel

**DOI:** 10.3390/cancers14215277

**Published:** 2022-10-27

**Authors:** Giulio Francolini, Ilaria Morelli, Maria Grazia Carnevale, Roberta Grassi, Valerio Nardone, Mauro Loi, Marianna Valzano, Viola Salvestrini, Lorenzo Livi, Isacco Desideri

**Affiliations:** 1Radiation Oncology Unit, Azienda Ospedaliero-Universitaria Careggi, 50134 Florence, Italy; 2Department of Biomedical, Experimental and Clinical Sciences “Mario Serio”, University of Florence, 50134 Florence, Italy; 3Department of Precision Medicine, University of Campania “L. Vanvitelli”, 80138 Naples, Italy; 4Italian Society of Medical and Interventional Radiology (SIRM), SIRM Foundation, 20122 Milan, Italy

**Keywords:** prostate cancer, radiomics, multi-parametric magnetic resonance imaging, prostate imaging-reporting and data systems, prostate-specific membrane antigen PET/CT

## Abstract

**Simple Summary:**

This paper aims at showing the impact of novel imaging technologies and modern radiotherapy techniques on the management of cancer, with a particular focus on prostate adenocarcinoma. The manuscript explores the value of diagnostic imaging before treatment, the role of radiomics in predicting outcomes, the benefit of novel imaging in radiotherapy planning and the influence of advanced technologies in systemic treatment and in the management of other non-oncological conditions in order to tailor the best therapeutical strategies.

**Abstract:**

Introduction: Targeted dose-escalation and reduction of dose to adjacent organs at risk have been the main goal of radiotherapy in the last decade. Prostate cancer benefited the most from this process. In recent years, the development of Intensity Modulated Radiation Therapy (IMRT) and Stereotactic Body Radiotherapy (SBRT) radically changed clinical practice, also thanks to the availability of modern imaging techniques. The aim of this paper is to explore the relationship between diagnostic imaging and prostate cancer radiotherapy techniques. Materials and Methods: Aiming to provide an overview of the integration between modern imaging and radiotherapy techniques, we performed a non-systematic search of papers exploring the predictive value of imaging before treatment, the role of radiomics in predicting treatment outcomes, implementation of novel imaging in RT planning and influence of imaging integration on use of RT in current clinical practice. Three independent authors (GF, IM and ID) performed an independent review focusing on these issues. Key references were derived from a PubMed query. Hand searching and clinicaltrials.gov were also used, and grey literature was searched for further papers of interest. The final choice of papers included was discussed between all co-authors. Results: This paper contains a narrative report and a critical discussion of the role of new modern techniques in predicting outcomes before treatment, in radiotherapy planning and in the integration with systemic therapy in the management of prostate cancer. Also, the role of radiomics in a tailored treatment approach is explored. Conclusions: Integration between diagnostic imaging and radiotherapy is of great importance for the modern treatment of prostate cancer. Future clinical trials should be aimed at exploring the real clinical benefit of complex workflows in clinical practice.

## 1. Background

Improvement of clinical outcomes through targeted dose-escalation and reduction of dose to adjacent organs at risk has been the main goal of radiotherapy in the last decade. Due to its characteristic radiobiology, prostate cancer benefited the most from this process, with a significant improvement in the benefit-to-risk ratio. Implementation of three-dimensional conformal planning (3-d CRT) was the first step of this evolution, followed by the introduction of Cone-Beam Computed Tomography (CBCT) for set-up evaluation. In recent years, the development of Intensity Modulated Radiation Therapy (IMRT) and Stereotactic Body Radiotherapy (SBRT) techniques radically changed clinical practice and allowed us to re-consider the role of radiotherapy in different clinical scenarios. However, all these upgrades were made possible by the availability of modern imaging techniques, both for their diagnostic value and for the possibility of integrating their information with standard radiotherapy planning. In this paper, our purpose was to overview the close relationship between diagnostic imaging and prostate cancer radiotherapy techniques development.

## 2. Diagnostic Imaging and Its Predictive Value before Treatment: How Modern Imaging May Avoid Unnecessary Invasive Diagnostic Procedures (Table 1 and Table 2 Below)

Similarly to other tumours [1,2,3,4], multiparametric magnetic resonance imaging (mpMRI) has been proposed as a widely adopted standard for diagnosis and baseline assessment of prostate cancer [5]. For example, Prostate Imaging Reporting and Data System (PI-RADS) represents one of the most important efforts performed in recent years to standardize mpMRI reporting [6]. However, despite the improvements in terms of implementation of this imaging method in clinical practice, it is well known that some issues still exist, like the management of Prostate Imaging Reporting and Data System (PI-RADS) score 3 lesions or the inter-reader variability among clinicians with different levels of expertise [7]. Moreover, various essays have been tested to improve the accuracy of mpMRI in this setting [8]. Indeed, many clinicians feel that the PI-RADS v2 scoring system can be inadequate in distinguishing clinically significant and insignificant groups in central gland tumours [9]. Moreover, it is important that radiologists are familiar with the common incidental findings associated with MRI to minimize the anxiety of the patient and to reduce costs associated with unnecessary further testing [10] since it is currently acknowledged that the widespread request for mpMRI significantly increased the identification of findings unrelated to the primary aim of the investigation [11]. Nonetheless, mpMRI confirmed its clinical usefulness by predicting prostate cancer Gleason Grade before biopsy [12,13] and by detecting local recurrence after radical prostatectomy [14,15]. All these features are of utmost clinical importance in planning either definitive or postoperative prostate radiotherapy.

**Table 1 cancers-14-05277-t001:** Diagnostic imaging and predictive value before cancer treatment in other diseases.

Study	Year	N. of Pts	Type of Cancer	Diagnostic Imaging	Primary Endpoint	Main Findings
Pietragalla M et al. [1]	2020	92	Salivary glands	MRI	Predictive role of ADC	ADC can be used as a parameter of benignity and malignancy in salivary gland neoplasms
Srisajjakul S et al. [2]	2020	-	Pancreatic cancer	CT and MRI	To distinguish between chronic pancreatitis and pancreatic cancer by means of CT and MRI features	Combinations of several imaging signs and features can improve accurate differentiation between pancreatitis and cancer
Granata V et al. [3]	2021	88	Cholangiocarcinoma	MRI	To analyze the features of ICC and its differential diagnosis at MRI	MRI features allowed the differentiation between mass-forming ICCs and other mimickers with statistical significance
Mungai F et al. [4]	2021	-	Salivary glands	DCE-MRI	To evaluate DCE-MRI parameters as imaging biomarkers for characterization and differentiation between benign and malignant lesions	DCE-MRI pharmacokinetic data could be helpful for recognizing the principal types of salivary gland tumors

MRI, magnetic resonance imaging; ADC, apparent diffusion coefficient; CT, computed tomography; ICC, intrahepatic cholangio carcinoma; DCE-MRI, diffusion contrast-enhanced magnetic resonance imaging.

**Table 2 cancers-14-05277-t002:** Diagnostic imaging and predictive value before cancer treatment in prostate cancer.

Study	Year	N. of Pts	Type of Cancer	Diagnostic Imaging	Primary Endpoint	Main Findings
El-Shater Bosaily A et al. [5]	2015	714	Prostate cancer	MP-MRI	MP-MRI diagnostic role	MP-MRI of the prostate prior to the first biopsy improves the detection accuracy of clinically significant cancer
Turkbey B et al. [6]	2019	-	Prostate cancer	MRI	PI-RADS v2.1 modifications and better assessment of prostate cancer on MRI	The updated version PI-RADS v2.1 can improve inter-reader variability and simplify the assessment of prostate cancer on MRI
Scialpi M et al. [7]	2021	-	Prostate cancer	MRI	To increase the accuracy of MP-MRI to reduce equivocal lesions and unnecessary biopsies	A simplified PI-RADS (S-PI-RADS) is an easy, reliable and potentially reproducible system for detecting and managing prostate cancer
Morote J et al. [8]	2022	567	Prostate cancer	MRI	To identify candidates for prostatic biopsy among patients in the PI-RADS 3 category	The Proclarix test is more accurate in selecting appropriate candidates for prostate biopsy among men in the PI-RADS 3 category
Gundogdu E et al. [9]	2020	-	Prostate cancer	MRI	Relationship between serum PSA level, GS, PI-RADS v2 score, ADCmin value and tumour diameter in patients who underwent RP	PI-RADS v2 scoring system, tumour diameter, tumour ADC_min_ values and PSA value can be predictive parameters in both central and peripheral carcinomas
Trivedi J et al. [10]	2021	-	Prostate cancer	MRI	To identify incidental findings in and around the prostate on MRI	Radiologists must be familiar with common incidental findings on MRI to minimise anxiety in the patient, have a well-informed discussion with the referring clinician and reduce costs and follow-up
Cutaia G et al. [11]	2020	647	Prostate cancer	MRI	To assess the prevalence and clinical significance of incidental findings on prostatic MP-MRI	Incidental findings might be encountered frequently on MP-MRI, and they are more common in patients aged > 65
Gong L et al. [12]	2022	489	Prostate cancer	MRI	To assess the correlation between prostate gland radiomic features and GS	2D prostate-gland-MRI-based radiomic features showed stable potential in identifying GS
Santone A et al. [13]	2021	112	Prostate cancer	MRI	To detect prostate cancer grade by radiomic features directly from magnetic resonance images	Effectiveness of radiomics for Gleason grade group detection from magnetic resonance
Renard-Penna R et al. [14]	2022	-	Prostate cancer	MRI	To assess the role and potential impact of MRI in targeting local recurrence after surgery for prostate cancer in the setting of salvage radiation therapy	Functional MRI with diffusion and perfusion imaging has the potential to demonstrate local recurrence even at low PSA values
Coppola A et al. [15]	2020	73	Prostata cancer	CE-MRI	To evaluate the role of CE-MR in the diagnosis of local recurrence in patients with prostate cancer after radical prostatectomy and referred for salvage radiotherapy	The sensitivity of CE-MRI in local recurrence detection after radical prostatectomy increases with increasing PSA values

MP-MRI, multi parametric magnetic resonance imaging; PIRADS, prostate imaging reporting and data systems; GS, gleason score.

## 3. Radiomics: Role of Imaging in Predicting Treatment Outcomes: How Modern Imaging May Help to Predict Outcomes of a Determined Treatment (Table 3 and Table 4 Below)

Integration between modern imaging and radiomics may also be helpful for diagnosis [16,17,18,19,20] and for predicting radiation toxicities before treatment [21]. Of course, treatment toxicity is one of the most important factors to consider to evaluate the benefit-to-risk ratio and to decide treatment strategy in modern radiotherapy [22], but radiological features may also be helpful in predicting clinical outcomes after definitive approaches for different neoplastic diseases [23,24,25,26,27,28,29,30,31]. Moreover, radiomics models were used to predict tumour histopathological features in different series [32,33,34,35,36,37,38]. Abdollahi et al. described a machine learning approach to explore rectal, bladder and femoral head toxicity, aimed at identifying predictive factors on baseline MRI images [39,40,41]. Radiomics features were also evaluated in a secondary analysis of the HYPRO trial, with a significant improvement in rectal bleeding predictability if compared to Dose Volume Histogram alone [42]. Early variations of the rectal wall were observed in patients undergoing RT on an MRI linear accelerator (MRI Linac) in the second week of treatment [43]. Radiomics features could become a valuable tool in this scenario, and machine learning may further refine treatment tailoring [44].

**Table 3 cancers-14-05277-t003:** Radiomics: role of imaging in predicting treatment outcomes in other diseases.

Study	Year	N. of Pts	Type of Cancer	Diagnostic Imaging	Primary Endpoint	Main Findings
Zhang L et al. [16]	2020	360	GIST	CT	CT-based radiomics models for GIST risk stratification	Quantitative radiomics analysis can be regarded as a complementary tool to achieve an accurate diagnosis for GISTs
Kirienko M et al. [17]	2020	108	Thymic neoplasms and lymphoma	CT	CT-based radiomics model in the differential diagnosis between lymphomas and thymic neoplasms	Radiomics analysis support diagnosis in patients with mediastinal masses with a major impact on the management of asymptomatic cases
Lian S et al. [18]	2020	147	NPC and NPL	MRI	Predictive role of ADC in differentiating NPC from NPL at the primary site	Whole-tumour histogram analysis of ADC maps could be helpful for differentiating NPC from NPL
Nakamura Y et al. [19]	2021	-	HCC	CT	Role of advanced CT techniques in assessing HCC	Dual-energy CT, perfusion CT and AI-based methods can be used for the characterization of liver tumours, the quantification of treatment responses, and for predicting the overall survival rate of patients
Danti G et al. [20]	2021	-	NEN	CT and ^68^Ga-DOTA-peptides PET/CT	Morphologic and functional imaging in the characterization of NEN	Potential future possibilities of prognostic imaging in the assessment of NEN, especially GI ones
Farchione A et al. [23]	2020	57	NSCLC	CT	Predictive role of CT data in NSCLC survival	Opposite influence on the performance of quantitative imaging features in predicting OS of surgically treated NSCLC patients
Sun NN et al. [24]	2020	72	Oesophageal cancer	DCE-MRI	To predict and assess treatment response by DCE-MRI in patients treated with CRT	DCE-MRI could serve as an imaging technique for treatment planning
Crimì F et al. [25]	2020	62	Rectal cancer	T2w-MRI	Role of T2-weighted MRI in the prediction of outcomes in patients LARC undergoing nCRT	MRI T2-weighted sequences-based TA was not effective in predicting the complete pathological response to nCRT in patients with LARC
Fornell-Perez R et al. [26]	2020	100	Rectal cancer	DWI-MRI	To assess the value DWI to HRT2w in MRI detection of EMVI	The addition of DWI improved the diagnostic performance of EMVI
Pietragalla M et al. [27]	2020	40	Laryngeal carcinoma	CT	To evaluate cartilage invasion on CT in patients undergoing total laryngectomy	CT has a high predictive value in assessing cartilage invasion in both primary and recurrent carcinomas
Russo L et al. [28]	2021	13	Cervical cancer	MRI	Role of MRI in the evaluation of response to treatment after nCRT in patients with cervical cancer FIGO stage IB2-IIA1	The usefulness of MRI in the assessment of treatment responses after NACT
Hang-Tong H et al. [29]	2020	-	HCC	CT	CT image-based radiomics model for early recurrent HCC	CT-based radiomics has poor reproducibility between centres. Image heterogeneity, such as slice thickness, can be a significant influencing factor
Cusumano D et al. [30]	2021	195	Rectal cancer	MRI	To develop a generalised radiomics model for predicting pCR after nCRT in LARC patients	Good performance of the elaborated model
Hu S et al. [31]	2020	225	Papillary thyroid carcinoma	MRI and US	Role of US and MRI in predicting ETE in patients with papillary carcinoma	Preoperative US should be used as the first line in predicting minimal ETE, and MRI should be added to extensive ETE assessment
Zhang Y et al. [32]	2020	128	Breast carcinoma	MRI	Role of MRI in predicting Ki67 index	The ADC-based radiomics model is a feasible predictor for the Ki-67 index in patients with invasive ductal breast cancer
Nazari M et al. [33]	2020	71	Renal Carcinoma	CT	CT image-based radiomics model for ccRCC grade	CT radiomics features a useful and promising methodology for the preoperative evaluation of ccRCC Fuhrman grades
Benedetti G et al. [34]	2021	39	Pancreatic cancer	CT	CT image-based radiomics model in detecting histopathologic characteristics of pancreatic NET	Radiomics features can discriminate histopathology of panNET
Halefoglu AM et al. [35]	2021	66	Renal carcinoma	CT and MRI	To investigate whether CT and T2 weighted- MRI could discriminate between low grade and high grade in ccRCC and pRCC	Contrast-enhanced CT and T2 weighted -MRI can play a considerable role in the discrimination of low-grade versus high-grade tumours of both subtype RCC patients
Danti G et al. [36]	2020	68	Lung carcinoid	CT and nuclear imaging	To assess CT and nuclear imaging’s role in characterizing lung carcinoid	CT and nuclear molecular imaging are important in characterizing lung carcinoids
Bracci S et al. [37]	2021	72	NSCLC	CT	CT image-based radiomics model in predicting PDL1 expression in advanced NSCLC	CT texture analysis could be useful for predicting PD-L1 expression and guiding the therapeutic choice in patients with advanced NSCLC
Agazzi G et al. [38]	2021	84	NSCLC	CT	Predictive role of CT texture-based model in detecting EGFR-mutational status and ALK rearrangement	Texture analysis could be promising for the noninvasive characterization of lung adenocarcinoma with respect to EGFR and ALK mutations
Desideri I et al. [44]	2020	-	Head and Neck, Breast, Lung and Prostate cancer	-	Current-state-of-the-art on the use of radiomics for the prediction of radiation-induced toxicity	The current state-of-the-art on radiomics prediction of radiation-induced toxicity is still relatively limited, with the notable exception of xerostomia prognostication

GIST, gastro-intestinal stromal tumour; CT, computed tomography; NPC, nasopharyngeal carcinoma; NPL, nasopharyngeal lymphoma; ADC, apparent diffusion coefficient; HCC, hepatocellular carcinoma; AI, artificial intelligence; NEN, neuroendocrinal neoplasms; NSCLC, non-small cell lung carcinoma; TA, texture analysis; DCE-MRI, diffusion contrast-enhanced MRI; LARC, locally advanced rectal cancer; nCRT, neoadjuvant chemo-radiotherapy; pCR, pathological complete response; ETE, extra-thyroid extension; ccRCC, clear-cell renal cell carcinoma; pRCC, papillary renal cell carcinoma; IMRT, intensity-modulated radiation therapy.

**Table 4 cancers-14-05277-t004:** Radiomics: role of imaging in predicting treatment outcomes in prostate cancer.

Study	Year	N. of Pts	Type of Cancer	Diagnostic Imaging	Primary Endpoint	Main Findings
Mostafaei S et al. [21]	2020	64	Prostate cancer	CT	Prediction of RT-induced toxicity by means of CT radiomics	CT imaging features can predict radiation toxicity in association with clinical and dosimetric features
Abdollahi H et al. [39]	2018	-	Prostate cancer	MRI	Role of MRI radiomic analysis to assess IMRT rectal toxicity	Pre-IMRT MR image radiomic features could predict rectal toxicity in prostate cancer patients
Abdollahi H et al. [40]	2019	33	Prostate cancer	MRI	Role of MRI texture features analysis in predicting IMRT urinary toxicity	Radiomics features as potentially important imaging biomarkers in radiation-induced bladder injuries
Abdollahi H et al. [41]	2019	30	Prostate cancer	MRI	Role of MRI texture features analysis in predicting IMRT femoral head damage	Radiomics features as potentially important imaging biomarkers for predicting radiotherapy-induced femoral changes
Rossi L et al. [42]	2018	351	Prostate cancer	-	Role of TA 3D distributions in predicting toxicity rates	3D dosimetric texture analysis features have a predictive role in detecting GI and GU radiation toxicity
Lorenz JW et al. [43]	2019	-	Prostate cancer	MRI-Linac	MRI-Linac radiomics feature variation in OARs	Early variations of the rectal wall were observed in patients undergoing RT on MRI linear accelerator

## 4. Implementation of Novel Imaging in Radiotherapy Planning: How Modern Imaging May Improve Planning Radiotherapy Techniques (Table 5 and Table 6 Below)

Novel metabolic imaging improved staging sensibility in prostate cancer and has been shown to have a significant impact on radiotherapy management [45,46]. However, in the modern era, imaging influences radiotherapy planning in a deeper meaning. Since the introduction of three-dimensional imaging and planning, technical innovation has allowed us to increase the radiotherapy dose delivered to the target and to spare at the same time adjacent organs at risk. This trend was practice-changing, especially for prostate cancer, where a low alpha/beta ratio favours hypofractionated regimens [47] characterized by higher doses for a single fraction. Implementation of hypofractionated regimens in clinical practice is based on Image Guided Radiotherapy (IGRT), which has been used for the treatment of many different neoplastic diseases [48] and currently establishes a modern treatment standard, with favourable comparison to conventional treatment [49,50]. Lately, many reports about MRI-guided boosts to dominant intraprostatic lesions have been reported [51]. Moreover, modern treatment planning allowed us to push forward the ability to assess and report low radiation doses to organs at risk, thus helping to refine awareness about incidental irradiation of healthy tissue during treatment [52] and to explore the impact of dose to structures (e.g., bladder neck) on toxicity and quality of life [53]. Diagnostic imaging can be co-registered with planning imaging to drive modern high-dose intensity-modulated radiotherapy [54,55,56,57]. Kuisma et al. reported data about 30 men treated with radiotherapy and undergoing focal boost to Carbon-acetate PET/CT metabolically active areas. The authors concluded that this metabolic guidance for dose-escalated radiotherapy was feasible and deserved further study [58]. Moreover, Prostate Specific Membrane Antigen (PSMA) PET/CT-guided local ablative radiotherapy was tested in a prospective phase 2 trial enrolling oligometastatic prostate cancer patients treated on all PSMA-positive metastases without any systemic treatment. Briefly, this approach showed to be safe and effective in a selected population [59]. The final evolution of integration between imaging and radiotherapy is expressed in the use of online MRI to perform modern IGRT and adaptive RT with MRI Linac [60].

**Table 5 cancers-14-05277-t005:** Implementation of novel imaging in radiotherapy planning in other diseases.

Study	Year	N. of Pts	Type of Cancer	Diagnostic Imaging/Technique	Primary Endpoint	Main Findings
Rosa C et al. [48]	2021	32	Rectal cancer	CBCT	Role of CBCT in evaluating organ motion in LARC	CBCTs resulted in effective organ motion assessment, and it could be an appropriate method for the implementation of an intensification treatment

CBCT, Cone-Beam Computed Tomography; LARC, Locally Advanced Rectal Cancer.

**Table 6 cancers-14-05277-t006:** Implementation of novel imaging in radiotherapy planning in prostatic cancer.

Study	Year	N. of Pts	Type of Cancer	Diagnostic Imaging/Technique	Primary Endpoint	Main Findings
Jani AB et al. [45]	2021	165	Prostate cancer	^18^F-fluciclovine-PET/CT	Role of 18F-fluciclovine-PET/CT in post-prostatectomy radiotherapy decision-making and planning	The inclusion of ^18^F-fluciclovine-PET into postprostatectomy radiotherapy significantly improved survival free from biochemical recurrence or persistence
Hofman S et al. [46]	2020	302	Prostate cancer	PSMA PET/CT	Role of PSMA PET/TC in staging and management of HR prostate cancer	PSMA PET-CT is a suitable replacement for conventional imaging, providing superior accuracy to the combined findings of CT and bone scanning
Jereczek-Fossa B et al. [49]	2020	179	Prostate Cancer	IGRT	Incidence and predictors for outcomes and toxicity with IGRT	IGRT allows for safe, moderate hypofractionation
Cuccia F et al. [50]	2020	170	Prostate cancer	HT	Toxicity and clinical outcomes of moderately hypofractionatedHT	Moderately hypofractionated RT with HT for localized prostate cancer reported excellent outcomes with mild acute and late toxicity incidence, with promising biochemical control rates
Kerkmeijer L et al. [51]	2021	571	Prostate cancer	EBRT	Role of focal boosting of the macroscopically visible tumour in increasing bDFS	The addition of a focal boost to the intraprostatic lesion improved bDFS for patients with localized intermediate- and high-risk prostate cancer without impacting toxicity and quality of life
Onal C et al. [52]	2020	40	Prostate cancer	VMAT	To compare testicular doses during VMAT in patients receiving prostate-only and pelvic lymphatic irradiation	The patients with prostate-only irradiation received lower testicular doses than those with additional pelvic field irradiation
Sanmamed N et al. [53]	2019	61	Prostate cancer	MRI-guided high-dose-rate brachytherapy + EBRT	Impact of the dose on bladder neck urinary toxicity and health-related quality of life	A high bladder-neck dose was observed in patients who had acute urinary toxicity, but the predictive value of this parameter needs further investigation
D’Angelillo RM et al. [54]	2020	150	Prostate cancer	18F-choline PET/CT	Role of 18F-choline PET/CT in allowing dose escalation in salvage radiotherapy	High-dose salvage radiotherapy to a biological target volume is feasible on 18F-choline PET/CT positive areas
Rigo M et al. [55]	2020	24	Prostate cancer	EBRT	Role of EBRT molecular-imaging guided in salvage radiotherapy post-HIFU	Feasibility and low toxicity of salvage EBRT after HIFU failure
Francolini G et al. [56]	2022	-	Prostate cancer	IMRT/VMAT	Clinical outcomes nodal IMRT boost	WPRT and IMRT/VMAT boost on positive pelvic nodes are effective and promising approaches with limited toxicity
Francolini G et al. [57]	2022	185	Prostate cancer	SRT/SSRT	To compare outcomes between SRT and SSRT after radical prostatectomy and macroscopic recurrence	SSRT should be considered an attractive alternative to conventional SRT in this setting
Kuisma A et al. [58]	2022	30	Prostate cancer	Carbon-11 acetate (ACE) PET-CT	Role of Carbon-11 acetate (ACE) PET-CT for RT planning	Biological guidance for dose-escalated prostate RT is feasible with ACE PET/CT
Holscher T et al. [59]	2022	63	Prostate cancer	PSMA PET/CT	Toxicity and efficacy of Local Ablative PET PSMA-guided RT in oligometastatic patients	Local ablative radiotherapy is safe, and it might be an option to avoid systemic therapy in selected patients
Mazzola R [60]	2021	20	Prostate cancer	PSMA PET/CT, SBRT, MRI-linac	Feasibility and patient-reported outcomes following PSMA-PET/CT guided SBRT by means of MRI-Linac	Encouraging findings in terms of effectiveness and tolerability

PET, positron emission tomography; CT, computed tomography; PSMA, prostate-specific membrane antigen; IGRT, image-guided radiation therapy; HT, helicoidal tomotherapy; EBRT, external beam radiation therapy; bDFS, biochemical disease-free survival; VMAT, volumetric modulated arc therapy; HIFU, high-intensity focused ultrasound; IMRT, intensity-modulated radiation therapy, WPRT, whole-pelvic radiation therapy; SRT, stereotactic radiation therapy; SSRT, stereotactic synchroton radiotherapy; SBRT, stereotactic body radiation therapy.

## 5. Influence on Systemic Treatment and Different Medical Conditions: How Modern Imaging May Change Use of Radiotherapy Influencing Overall Treatment Management of Patients (Table 7 Below)

Local ablative treatment has been shown to be a practice-changing approach in oligometastatic prostate cancer [61], and the integration of imaging techniques and modern radiotherapy allowed us to increase the potential areas of exploitation of radiotherapy as a non-invasive, short and profitable technique. For example, stereotactic body radiotherapy (SBRT) is currently used in routine clinical practice as an ablative approach, able to control loco-regional sites of oligo-progression, thus deferring change of systemic treatment [62,63] or even in non-neoplastic diseases, thus opening emerging clinical scenarios [64]. Local ablative treatment is important in prostate cancer management due to the observation that metastasis-to-metastasis spread is a common phenomenon for this disease [65]. This innovative approach prompted radical changes in terms of clinical management of prostate cancer for clinical oncologists, especially considering the rising necessity for developing integrated treatment strategies with Androgen Deprivation Therapy and new hormonal agents (Abiraterone, Enzalutamide and Apalutamide) [66]. Moreover, the COVID-19 pandemic prompted us to adapt patients’ management [67,68,69,70,71], and the availability of shorter techniques to enable us to deliver safe and effective radiotherapy treatment was of utmost importance during the COVID-19 pandemic [72,73,74]. 

**Table 7 cancers-14-05277-t007:** Influence on systemic treatment and different medical conditions.

Study	Year	N. of Pts	Type of Cancer	Diagnostic Imaging/Technique	Primary Endpoint	Main Findings
Phillips R et al. [61]	2020	54	Prostate cancer	SABR	To determine if SABR improves oncologic outcomes compared to observation	Treatment with SABR for oligometastatic prostate cancer improved outcomes
Vernaleone M et al. [62]	2019	38	Rectal cancer	SBRT	Safety and clinical benefit of SBRT for liver oligometastatic colorectal cancer	Importance of patients’ selection to identify the oligometastatic scenario most likely to benefit from SBRT. Prospective studies are needed to further assess its role among locoregional treatment options for liver metastases from CRC
Falcinelli L et al. [63]	2021	56	Lung cancer	SBRT	Outcomes after SBRT on primary and metastatic lung cancer	SBRT was well tolerated and provided good local control
Fiorentino A et al. [64]	2021	-	Cardiac arrhythmias	SABR	Role of SABR in treating cardiac arrhythmias	Clinical data suggested the feasibility, efficacy and safety of SABR for refractory ventricular arrhythmias
Barra S et al. [72]	2021	28	Prostate cancer	SBRT	To evaluate SBRT in low-risk prostate cancer patients as a treatment option in emergency health conditions	SBRT for early prostate cancer reported a safe toxicity profile and a good clinical outcome at the median follow-up of 5 years, and it could be a useful option if radiotherapy is required in emergency medical conditions

SABR, Stereotactic Ablative Radiotherapy; SBRT, Stereotactic Body Radiotherapy; CRC, Colorectal Cancer.

## 6. Discussion

Different types of diagnostic imaging are currently essential for prostate cancer diagnosis and treatment. Before radiotherapy delivery, mpMRI is widely used for diagnosis, and future development will allow the improvement of its diagnostic accuracy and to predict the biological aggressiveness of diseases even before a biopsy. This would allow an early and non-invasive assessment of disease in clinical practice. Moreover, early reports show that treatment toxicity after radiotherapy may be predicted by diagnostic imaging through radiomics features. During radiotherapy planning and delivery, three-dimensional imaging may allow the implementation of modern IGRT and refine the evaluation of dose distribution to healthy tissues. Moreover, information from diagnostic imaging may be integrated into radiotherapy planning to better tailor treatment (e.g., Simultaneous Integrated Boost to positive nodal disease detected with choline PET/CT) through co-registration with planning imaging. Finally, online MRI imaging allowed the development of MRI Linacs, able to safely deliver MRI-based IGRT with online adaptive protocols in order to shape radiotherapy planning on a daily basis, taking into account the inter- and intra-fraction variability of patients’ anatomy [75]. Limitations of these approaches are mainly related to three aspects: first, the sensitivity and specificity of novel diagnostic imaging (e.g., mp-MRI for prostate cancer) have to be further validated in clinical trials. Diagnostic imaging, with a particular focus on MRI, can play a crucial role in the distinction between benign and malignant conditions, thus avoiding the need for much more invasive approaches such as biopsies. In this regard, we cited Pietragalla et al. experience in 92 salivary gland patients, where diffusion-weighted imaging (DwI) and dynamic contrast-enhanced perfusion-weighted imaging (DCE-PwI) have proven useful in detecting benign neoplasms, epithelial malignancies, Warthin tumours and lymphomas [1]; distinguishing between benign chronic pancreatitis and pancreatic cancer by means of CT and MRI imaging was instead the main purpose of Srisajjakul and colleagues work [2]. Granata et al., always in the setting of gastrointestinal neoplasms, analyzed 88 patients and tried to outline peculiar MRI features which could allow the differentiation between intrahepatic cholangiocarcinoma and other forming-mass benign mimickers [3]. El-Shater Bosaily, Turkbey, Scialpi and Morote exploited the potential role of MRI diagnostic assessment in prostatic disease: they showed that the PI-RADS v2 scoring system could limit the detection of equivocal lesions, thus leading to the avoidance of unnecessary biopsies and to the accurate selection of patients who really deserve such an invasive diagnostic approach [5,6,7,8]. 

Among histologically confirmed tumours, new innovative technologies can provide better diagnostic refinement. As an example, Mungai and coll. showed how DCE MRI data can be helpful in recognizing the most frequent histopathological types of salivary neoplasms [4]; for what concerns prostatic disease, radiomic features can help in the process of risk-stratification: Gong et al., in their 489-prostate cancer patients experience, showed a correlation between radiomic features on mp-MRI and Gleason Score [12], as well as Santone et al. who demonstrated in a smaller sample size population the effectiveness of radiomics in grade group detection from MRI [13]. Functional MRI can also predict the risk of local recurrence after prostate surgery both at low PSA values, as demonstrated by Renard-Penna [14], and along with an increase in PSA with much more sensitivity, as in Coppola’s work [15]. 

As a consequence, in the pre-treatment setting, this is of particular interest when diagnostic imaging could be integrated with clinical data to differentiate between benign or neoplastic findings, avoiding unnecessary histopathologic assessment through biopsy. Clinical trial design to address this issue is complicated and will probably require a large shared dataset of patients with a longitudinal follow-up. Moreover, as diagnostic imaging rapidly evolves, such databases could become obsolete if compared to current clinical standards. Currently, evidence from literature often does not allow to safely rely only on imaging for oncologic diagnosis. Larger series have been published for prostate cancer, but the number of patients included in these experiences is still too low (e.g., <1000 patients even in the largest series) to draw firm conclusions. Second, radiomics may offer the possibility to predict adverse events before treatment, but the assessment of this opportunity needs analysis of large populations and datasets. In this setting, radiomic features may play a role in assessing survival and toxicity outcomes and in predicting treatment responses in different cancer populations. Among prostate cancer patients, many studies investigated the potential of radiomic signatures in predicting radiotherapy-induced gastrointestinal (GI) and genitourinary (GU) toxicities. Mostafei’s prospective work on 64 patients, whose pre-RT CT scans of the rectum and of the bladder were acquired and whose GI and GU toxicities were assessed, demonstrated that CT imaging features could predict radiation-induced toxicity and that integrating imaging and clinical/dosimetric features may enhance the predictive performance of radiotoxicity modelling [21]. Abdollahi explored RT-related bladder, rectal and femoral head injuries separately in patients who underwent IMRT for prostate cancer: in this study, 274 radiomic features were extracted from T2-weighted sequences, and changes from pre-RT and post-RT imaging showed a good correlation with radiation dose and radiation-induced urinary toxicity [40]; a similar work on prostate cancer patients treated with RT, and whose radiomic features on MRI T2-weighted scans were detected before and after treatment, resulting in the pre-IMRT predictive value of rectal toxicity [39]. Radiomic features can also predict femoral head injuries and the risk of post-treatment fractures in PCa patients. Abdollahi analyzed 30 patients and 60 femoral heads and extracted 34 features from T1, T2 and DWI MRI scans whose changes revealed useful in predicting the risk of fracture and with a potential role as biomarkers [41]. Lorenz conducted an exploratory study in which the analysis of delta-radiomic (variations in quantitative image metrics) profiles in organs at risk (OARs) revealed significant changes in the bladder and rectal wall just after 1 week of RT in prostate cancer patients whose MRI scans were acquired on an MRI linac [43]. A useful summary of available literature was provided by Desideri et al., who evaluated the current state of the art on radiomics prediction of radiation-induced toxicity in Head and Neck, Breast, Lung and Prostate cancer, with the conclusion that at present, radiomics is not readily applicable to patient management in clinical practice, but holds great potential for improved clinical decision making in precision radiation oncology [44]. 

For what concerns the prediction of survival as well, many series underlined the potential of radiomic features in NSCLC surgically treated patients [23], in hepatocellular carcinoma [19] and in neuroendocrine neoplasms [20]. 

Interestingly, in the setting of locally advanced rectal carcinomas treated with neoadjuvant chemo-radiation, texture analysis on T2-weighted MRI sequences was also investigated for the prediction of pathological response. If Crimì experience discouraged the use of radiomic features in the prediction of cPR in rectal cancer [25], on the other hand, Russo and colleagues proved the usefulness of MRI in the assessment of response in a setting of cervical cancer patients staged IB2-IIA1 [28]. 

The predictive function of radiomic features analysis for toxicity and treatment/survival outcomes is probably complicated by the fact that many radiomics features have been explored with different imaging methods, with significant heterogeneity in terms of aims and outcomes chosen for features of interest. Artificial intelligence will be needed to address this issue [76]. For these reasons, it is difficult to predict when radiomics features could be steadily implemented in clinical practice because standardization in this field is yet to come. The third issue regarding radiomic features addresses both radiotherapy planning and delivery. With a peculiar focus on prostate cancer, superior accuracy in both phases of the radiotherapy process has been related to the implementation of novel imaging technologies. Among these, we find new PET tracers, such as 18F-fluciclovine, PSMA, coline or Carbon-11-ACE as well, which play a part in decision-making and planning [45,46], promoting dose-escalation [54,58], and guiding local ablative radiotherapy, thus avoiding the need of immediate systemic therapy [59,60]. In the delivery phase, Image-Guided Radiation Therapy [IGRT] is crucial in improving the precision and the accuracy of treatment, as it increases the probability of tumour control with shorter treatment schedules. IGRT is nowadays the standard of care for all cancer types. In prostate cancer patients, for example, it allows for a safe, moderate hypofractionated regimen [49]; in addition, the guidance of molecular imaging was the central issue in Rigo and colleagues’ work on 24 prostate cancer patients undergoing salvage radiotherapy after HIFU with efficacy and low toxicity [55]. 

The use of Cone-Beam CT (CBCT) in locally advanced rectal carcinoma is effective for mesorectum organ motion assessment, and it is an appropriate method for treatment implementation [48]. The role of MRI-guidance was also explored in this context: we cite Sanmamed’s work, aimed at assessing the impact of doses to the bladder neck on the development of acute GU toxicity and on QoL in patients treated with MRI-guided brachytherapy and EBRT [53]. 

Paradoxically, the implementation of novel imaging in RT treatment planning is currently the most advanced field of research and, in our opinion, the most exploitable one. Indeed, RT technologies are often routinely available and are often already well implemented in clinical practice. Moreover, direct clinical outcomes and benefits are evaluable after treatment (e.g., survival outcomes, response to treatment, etc.), and trial design is more traditional if compared to other research fields discussed above. However, much literature evidence is limited to retrospective case series, and prospective evidence should be advocated because the real clinical advantages of all techniques should be thoroughly evaluated to select patients who may benefit from a certain radiotherapy approach. Interestingly, the relationship between modern imaging methods and radiotherapy may constitute a paradigmatic example of a research field in which it would be complicated to wait for pre-treatment evidence, and assessing clinical oncological benefit from a certain approach could be more effective and less demanding in terms of the number of patients included in trials. Thus, a tight relationship between radiologists and radiotherapists should be advocated both in routine clinical practice and trial design in order to improve knowledge and maximize clinical benefit from technical innovation. Lastly, we must mention the fact that local treatment implemented by the integration of modern technologies may have an impact on systemic therapy in both oncological and non-oncological conditions. In prostate cancer, complete metastatic ablation of oligometastatic disease with SABR and integrated PSMA-PET may provide an alternative to early initiation of androgen deprivation therapy [61]; SBRT can also be useful in early prostate cancer in particular conditions (such as covid19 pandemics) requiring shorter treatment time [72]. SBRT was successfully implemented in oligometastatic colorectal cancer patients [62] and in patients with lung disease, providing good local control with good tolerability [63]. For what concerns non-oncological settings, we cited Fiorentino and colleagues’ work on cardiac arrhythmias: clinical data have proven the feasibility, efficacy and safety of SABR (25 Gy in one session) for refractory ventricular arrhythmias [64].

## 7. Conclusions

Integration between diagnostic imaging and radiotherapy is of cornerstone importance for the modern treatment of prostate cancer (Figure 1), and many enticing innovations are awaited to further improve the therapeutic ratio of radiotherapy in this field. However, awareness of the close interdependence between these two specialities is useful to underline the strengths and limitations of current radiotherapy approaches. Future clinical trial design should be aimed at exploring the real clinical benefit of complex workflows (e.g., diagnostic imaging, co-registration with planning imaging, dosimetric evaluation and online set-up correction) in clinical practice. Some recent advances in metabolic imaging may significantly improve staging in selected settings (e.g., PSMA-PET for prostate cancer disease), especially considering the detection of oligometastatic disease in patients candidates for SBRT [77]. However, many gaps still remain to be filled, radiomic and genomic biomarkers may further improve the predictive capacities of imaging methods, and the rise of deep learning techniques could increase synergism between modern imaging and radiotherapy [78].

## Figures and Tables

**Figure 1 cancers-14-05277-f001:**
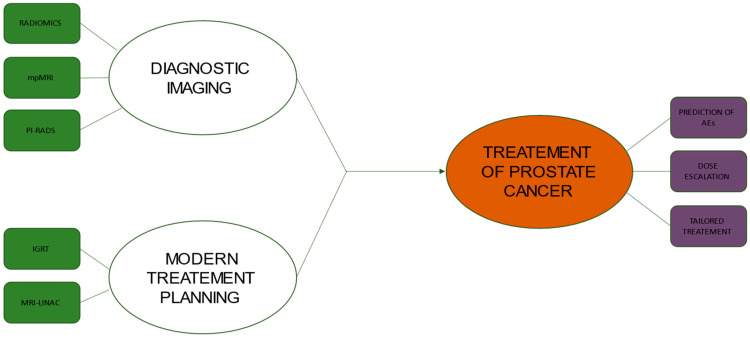
Flowchart showing the relationship between new diagnostic imaging technologies and modern treatment planning in the management of prostate cancer. ABBREVIATIONS: mpMRI, multi-parametric Magnetic Resonance Imaging; PI-RADS, Prostate Imaging–Reporting and Data System; IGRT, Image-Guided Radiation Therapy; MRI-LINAC, Magnetic Resonance Imaging Guided Linear Accelerator; AE, Adverse Events.

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
