# Peer review of "Integration between Novel Imaging Technologies and Modern Radiotherapy Techniques: How the Eye Drove the Chisel"

_cancers, 2022, doi:10.3390/cancers14215277_

Round 1

Reviewer 1 Report (Previous Reviewer 2)

The authors answered all comments and suggestions.

Author Response

We thank the Academic Editor for his suggestions and we here provide a point-by-point response to his advice:
1) We modified the tables 1 to 3 to as articles "related to prostate" and "other disease";
2) We added a dedicated paragraph in the "Materials and Methods" section in order to provide the selection criteria of the included articles;
3) We added specific subheadings for each review paragraph

This manuscript is a resubmission of an earlier submission. The following is a list of the peer review reports and author responses from that submission.

Round 1

Reviewer 1 Report

The subject has been discussed very superficially.  Each paragraph briefly summarizes the well- known news. There are no conclusions drawn after an in-depth discussion of the cited works. 

Author Response

We added some paragraphs to the discussion section in order to discuss in detail the cited works. 

Reviewer 2 Report

Prostate Cancer (PC) is the second most common cancer in men, worldwide. Multiparametric MRI (mpMRI) become a pivotal investigation to identify suspicious areas that are worthy to be biopsied. To standardize the interpretation of mpMRI, Prostate Imaging Reporting and Data System (PI-RADS) was born. Several technologies could be applied to manage PC such as radiomics, radiotherapies and/or systemic treatmens. One of PC concern is to predict the outcome before treatment. Indeed, the aim of this paper is to discuss critically the role of new modern techniques in predicting outcomes before treatment, in radiotherapy planning and in the integration with systemic therapy in the management of prostate cancer.

Comment to the Authors

Authors should be congratulated for the consolidative work. The evidence of the development of new integrated techniques represents a starting point to better manage PC patients and to guarantee a good improvement of survival outcomes. The manuscript is easily readable. The article adds to the current literature a lacking point of view for missing of:

  1. The discussion is lacking worthy novel key points such as these two novel papers (PMID: 34576134) on the role of new kind of integrated imaging to predict prognosis in PC. Authors should integrate the aspects discussed by these works.
  2. Moreover, this work (PMID: 34046207) on Ga prostate-specific membrane antigen PET-CT would enrich this work with a new perspective on the management of PC.

Author Response

We added a comment with appropriate referencing in the conclusion section. 

Round 2

Reviewer 2 Report

Tha authors answered all comments and suggestions.